# Behavioral phenotyping of mice lacking the deubiquitinase USP2

**Shashank Bangalore Srikanta**[1,2], **Katarina Stojkovic**[1,2], **Nicolas Cermakian**[2,3]*

**1** Integrated Program in Neuroscience, McGill University, Montréal, Québec, Canada, **2** Laboratory of Molecular Chronobiology, Douglas Research Centre, Montréal, Québec, Canada, **3** Department of Psychiatry, McGill University, Montréal, Québec, Canada

* nicolas.cermakian@mcgill.ca

**Data Availability Statement:** All data is available via the Figshare website. It can be found at https://figshare.com/s/39bed899c599fa965909.

**Funding:** This work was supported by grants from the Natural Sciences and Engineering Research Council of Canada (RGPIN-249731-2006 and

## Abstract

Ubiquitin specific peptidase 2 (USP2) is a deubiquitinating enzyme expressed almost ubiquitously in the body, including in multiple brain regions. We previously showed that mice lacking USP2 present altered locomotor activity rhythms and response of the clock to light. However, the possible implication of USP2 in regulating other behaviors has yet to be tested. To address this, we ran a battery of behavioral tests on *Usp2* KO mice. Firstly, we confirmed our prior findings of increased daily activity and reduced activity fragmentation in *Usp2* KO mice. Further, mice lacking USP2 showed impaired motor coordination and equilibrium, a decrease in anxiety-like behavior, a deficit in working memory and in sensorimotor gating. On the other hand, no effects of *Usp2* gene deletion were found on spatial memory. Hence, our data uncover the implication of USP2 in different behaviors and expands the range of the known functions of this deubiquitinase.

## Introduction

Ubiquitination is the process of covalent attachment of a 76 kDa long protein called Ubiquitin to other proteins. While ubiquitination has multiple roles in the functioning of the cell, including protein localization and trafficking [1], its most salient function is the targeting of tagged proteins for degradation [2]. Ubiquitination is carried out by a group of enzymes known as E3 ubiquitin ligases [2, 3]. The process of ubiquitination is counteracted by the process of deubiquitination, which is mediated by enzymes known as deubiquitinases or DUBs. These opposing processes can, in essence, determine the timing of various processes such as degradation, activity and localization of proteins.

While various DUBs are found in different tissues and cell types, Ubiquitin Specific Peptidase 2 or USP2 is one of the DUBs that has been found to be expressed almost ubiquitously in the organism [4–7]. More interestingly, it is the only DUB that has been found to have a circadian pattern of expression in virtually all the tissues where it is expressed [7, 8].

Circadian rhythms are self-sustaining oscillations with a period of approximately 24 hours, which are sustained at the molecular level by negative feedback loops involving circadian clock genes [9]. The counterbalance between ubiquitination and deubiquitination is essential to accurately control the timing of degradation of several clock proteins and hence, to the

RGPIN-2017-04675, to N.C.) and Velux Stiftung (Project 927, to N.C.), a graduate fellowship from the Faculty of Medicine, McGill University (to K.S.) and a graduate research stipend from the NSERC CD-CREATE Program (to S.S.). The funders had no role in study design, data collection and analysis, decision to publish, or preparation of the manuscript.

**Competing interests:** The authors have declared that no competing interests exist.

maintenance of periodicity in the clock [10]. Previous studies have shown that the deubiquitinase USP2 modulates the stability and/or localization of clock gene products BMAL1, PER1 and CRY1 [11–14]. Surprisingly though, despite its interaction with many of the core clock components, knocking out *Usp2* in mice did not lead to proportionally severe circadian deficits [11, 12]. Our group showed that USP2 modulates the response of the clock to light [11]. We also showed that *Usp2* KO mice have a slightly longer period than WT littermates [11]. While no other significant phenotypes were noticed under a normal light:dark cycle, a trend towards increased total activity over 24 hours was observed in the *Usp2* KO mice, compared to WT mice [11].

Ubiquitination, and ubiquitinating/deubiquitinating enzymes, have been associated with various behaviors as well as neurological and psychiatric disorders, such as bipolar disorder [15, 16], neurodevelopmental disorders [15, 17, 18], and Parkinson's disease [19]. Searching the mouse brain expression data in the Allen Mouse Brain Atlas [20], we noticed that *Usp2* is expressed in many brain regions. Hence, we aimed to test the hypothesis that USP2 plays a role in modulating various behaviors beyond circadian rhythms. To test this, we subjected *Usp2* KO and WT mice to a battery of tests, to assay various behaviors such as daily wheel running behavior, motor coordination, anxiety-like behavior, sensorimotor gating and memory. We confirmed that the *Usp2* KO mice showed increased and more consolidated running wheel activity, which was not due to increased motor coordination. Further, based on high *Usp2* expression in the hippocampus and cortex, we tested the mice for anxiety-like behavior and memory, and noticed a reduction in anxiety-like phenotypes and deficits in the working memory of *Usp2* KO mice, but not in their spatial memory. Based on *Usp2* expression in the olfactory bulb, striatum, thalamus and prefontal cortex, we assessed sensorimotor gating, for which limited effects of *Usp2* gene deletion were found.

## Materials and methods

### Animals

*Usp2* KO mice [21] were obtained from Dr. Simon S. Wing and bred in house. *Usp2* KO and WT littermates (on a C57BL/6J background) were generated by breeding heterozygotes. In some cases, PER2::LUC knockin mice (on a C57BL/6J background) were used as controls. This PER2 gene modification was shown not to affect PER2 function, mouse health, general activity or wheel-running activity [22]. Four such mice were used in the wheel running, rotarod, novel object recognition and pre-pulse inhibition tests. In all cases, similar effects of the KO were seen irrespective of whether or not these mice were included in the analyses.

Mice were weaned at 3 weeks of age, into cages containing no more than 5 mice per cage. They were maintained on a 12h:12h light:dark cycle (with light at ~200 lux). At 2 to 3 months of age, male mice were designated to running wheel or (non-circadian) behavioral cohorts. All procedures involving animals were carried out in accordance with guidelines of the Canadian Council on Animal Care and approved by the Animal Care Committees of McGill University and Douglas Mental Health University Institute (protocol no. 2001–4586). For euthanasia at the end of the experiments, the animals were first anesthetized by isoflurane inhalation, followed by $CO_2$ euthanasia; all efforts were made to minimize suffering.

### Wheel-running behavior

For running wheel experiments, 2 to 3-month-old WT and KO mice (n = 6) were transferred to running wheel-equipped cages (Actimetrics, Wilmette, IL, USA), where they were singly housed, with ad-libitum access to food and water. After a baseline week for acclimatization, wheel running activity was recorded for three weeks, in the same LD cycle. The last

10 days of recorded data were analysed on the Clocklab software Version 6 (Actimetrics, Wilmette, IL, USA). Daily wheel rotations, number of bouts of activity per day, length of each activity bout and number of wheel rotations in each bout were quantified. One mouse (KO) was excluded from the analysis because its wheel running activity was not recorded on the software.

## Non-circadian behavioral tests

WT mice (6-12/group) and *Usp2* KO mice (10-12/group) went through Rotarod, Elevated plus maze (EPM), Novelty suppressed feeding (NSF), Novel object recognition (NOR), Morris water maze (MWM) and Pre-pulse inhibition (PPI) of acoustic startle. To account for time-of-day effects on behavior, all behavioral tests were carried out between one and five hours after lights on. Tests were performed under dim light conditions (~15 lux) unless noted otherwise. The groups (WT and KO) were counterbalanced between tests and between trials. Mice were habituated to the testing room for 30 minutes, prior to the start of each session of behavioral testing. EPM, NSF and NOR tests were video recorded to minimize interaction with the mice and to reduce the resulting stress levels. All the mice in the video recordings of the behaviors were tracked using the TopScan 2.0 software (Clever Sys, Restin, VA, USA). Since some tests measured anxiety-like behaviors or could be affected by stress, all mice were given 2 "recovery days" between each test, during which they were not handled or disturbed.

**Rotarod.** The Rotarod test assays motor coordination, equilibrium and balance [23]. This test was carried out over two days. On the first day, each mouse was habituated to the rotarod (Bioseb, Valbonne, France). For the habituation phase, the ridged cylinder was made to rotate at a constant speed of 5 rotations-per-minute (rpm) for two 3-minute long trials and 10 rpm for two 3-minute long trials. During each of these trials, the mice were put back on the rod if they fell, until the timer ran out. Each mouse got a rest period of at least 15 minutes in its home cage between each subsequent trial.

For the testing day, each mouse was tested for three trials. It was placed on the rotarod, which accelerated at a constant rate, from 4 to 40 rpm, over a 5-minute span (acceleration = 7.2 rpmpm). When the mouse fell off the cylinder, onto the press plate placed 20 cm below the rod, the time taken to fall off the rod was recorded. After the fall, they were returned to their home cage. The greater their latency to fall, the greater is their motor coordination, equilibrium and/or balance. Two mice were excluded from this analysis (1 WT and 1 KO) as they refused to run on the rotarod.

**Open field test.** The open field test measures general locomotor activity in mice [24]. For this test, we used a VersaMax Legacy Open Field setup (AccuScan Instruments, Inc., Columbus, OH, USA). Mice were left to explore the VersaMax acrylic activity chamber with infrared sensors for 50 minutes while the experimenter was absent from the room. Data were collected using the Versamax Software (version 4.0, 2004; AccuScan Instruments, Inc., Columbus, OH, USA). From the collected data, total distance travelled in 50 minutes was plotted.

**Elevated plus maze.** Anxiety-like behavior was measured using the elevated plus maze (EPM) [25]. The maze is elevated 75 cm from the floor and consists of 4 arms shaped like a cross and painted black. Two opposing arms are enclosed by 10 cm high walls on three sides, while the other two are open. For the assay, each mouse was placed in the center of the maze, facing an open arm. The mouse was then allowed to freely explore the maze for 5 minutes, after which it was returned to its home cage. The total time spent in the open and closed arms, as well as in the center were measured. The longer a mouse spends in the open arms, the less anxiety-like behavior it is considered to show. Proportion of time spent in open arms was

calculated as:

$$\% \ time \ in \ open \ arms = \frac{(time \ spent \ in \ open \ arms(s)) + 0.5 * (time \ spent \ in \ the \ middle(s))}{300 \ s}$$

**Novelty-suppressed feeding.** The novelty suppressed feeding (NSF) test is an assay of anxiety-like behavior in a conflict-based environment [26, 27]. Mice were fasted for 24 hours before the start of the test. Then, each mouse was placed in a grey 48 cm X 48 cm X 48 cm box, in a well-lit area. At the center of the box, a fixed food pellet (standard chow) was available for the mouse to feed on. Thus, the mouse needs to choose between the anxiogenic setting of feeding in an open, well lit area as opposed to the safety of walls, but without any food availability. The mouse was allowed to explore this setting for 10 minutes and the latency to feed was measured, being careful to not include food manipulation or sniffing behaviors in this measure. The longer a mouse takes to feed, the higher its anxiety-like phenotype.

After this, the mouse was transferred to a quiet area, into a cage containing pre-weighed quantities of food and the amount of food they ate over a further 10-minute span was recorded. This measure assesses the hunger levels of the mice and whether differences due to the food deprivation itself influences the observations within the arena.

Additionally, to factor in the possibility of a general difference in appetite between KOs and WTs, at the end of the NSF protocol, the mice were weighed daily for 9 days and the amount of food that they consumed was noted over these days.

**Novel object recognition.** The novel object recognition (NOR) test measures learning and memory in mice [28]. NOR involves three phases spread over two days. On the first day, each mouse was acclimatized to an empty grey 48cm X 48cm X 48cm box by allowing it to freely explore the box for 10 minutes. On the second day, the first phase of the day is the habituation phase where the mouse was allowed to freely explore the grey box for 10 minutes. Each box contained two identical objects (either two 75 mm tissue culture flasks filled with corn cob bedding, or two boxes of coverslips) placed at opposite corners of the box, 12 cm away from each of the two nearest walls. The mouse was then returned to its home cage. The second phase of the day is the novel object recognition phase, which tests recall from working memory. In this phase, which starts 4 hours after the habituation phase, one of the objects in each box was exchanged for a differently shaped object (tissue culture flask as a replacement for coverslip box and vice-versa). The mouse was then allowed to freely explore the arena for 10 more minutes. The time spent exploring the two objects was separately recoded, in both phases. A mouse was considered to be involved in object exploration if its head was directed towards the object, within a radius of approximately 2 to 3 cm from it. This analysis was carried out using the TopScan 2.0 software. Mice tend to prefer exploring novel objects, over familiar objects. Hence, the exploration of the novel object is a measure of the extent to which the animal remembers the previous encounter with the familiar object. The ability to discriminate between the novel and familiar object was measured by the Discrimination Ratio (DR), calculated as:

$$DR = \frac{Time \ spent \ exploring \ novel \ object}{Time \ spent \ exploring \ familiar \ object}$$

A DR of 1 shows a lack of discrimination between the two objects. DR > 1 shows that the mouse is interacting with the novel object more than the familiar object.

Using the TopScan 2.0 software, the total distance traveled by the mice during the trials were also measured.

**Morris water maze.** The Morris water maze (MWM) assesses spatial memory [29, 30]. The maze consists of a large circular swimming pool (diameter of 150 cm) filled with opaque water with a platform (225 cm$^2$) submerged 1 cm below the surface of the water. The mice must reach the platform in order to be able to stop swimming and rest. Spatial cues were placed on the pool walls to allow the mice to learn the location of the platform. The learning stage was 4 days long, with each day consisting of 4 trials, each with the mouse beginning at a different location in the pool. The inter-trial interval was at least 30 min. The time to reach the platform was recorded at every trial, to compare the patterns of learning. On day 5 of the experiment, the platform was removed, and the time spent in each quadrant of the pool in the span of 1 minute was recorded, to test the robustness of spatial learning (probe trial). Then, a cue trial, where the platform is placed in a new location along with a visual cue, was administered to verify that the visual system was intact in the animals. The latency to reach the platform and time spent in the appropriate quadrant in the probe trial were analyzed and calculated using HVS Image Analysis (HVS Image, Hampton, UK).

**Pre-pulse inhibition of acoustic startle.** Pre-pulse inhibition (PPI) of acoustic startle reflex is a measure of sensory-motor gating [31, 32]. Testing is carried out using the SR-Lab software connected to 6 sound attenuating chambers equipped with plexiglass animal enclosure tubes (San Diego Systems, San Diego, CA, USA). These chambers are ventilated by an electrical fan that produces a constant 70 dB background. Speakers positioned directly above the enclosure present tone pulses of differing loudness and the startle of the animal is recorded by an accelerometer attached to the base of the enclosure. Following a 5-minute acclimatization period in the tube, there were 3 phases in the paradigm. In the first and third phases, 6 startle pulses of 120 dB loudness and lasting for 30 msec each, were administered. In the second phase, 38 trials were administered. The first 8 trials were pulse only (startle only) trials. In the next 30 trials, the mouse received a 30 msec pre-pulse of 0 (pulse alone), 6, 9, 12 or 15 dB intensity above the background, 100 msec prior to the actual 120 dB pulse. These pre-pulses were randomly varied across the 30 trials spaced 15 msec apart, 5 of each of the pre-pulse trials presented to the animals. The average amplitude of startle in the last 15 startle-only trials is the baseline startle. % Pre-pulse inhibition (% PPI) is calculated as:

$$\% \ PPI = 100 - \frac{Startle \ response \ to \ trials \ with \ pre-pulse}{Startle \ response \ to \ pulse \ alone \ trials} * 100$$

The higher the % PPI, greater is the inhibition of acoustic startle when subjected to a low intensity sound pulse prior to the startle. In essence, a higher PPI shows higher sensorimotor gating. Two mice were excluded from this analysis (1 WT and 1 KO) for having impossibly high baseline startle levels, pointing to a recording error in the data of these mice.

## Statistical analysis of data

All the data were analyzed first for homogeneity of variances as well as normality of data. The data were plotted and analyzed on GraphPad Prism, using the appropriate statistical tests. For all the tests with data distributed normally and having equal variance across the groups (WT, KO), unpaired two-tailed t-tests were used (Wheel running data, Rotarod, EPM, NSF, NOR, probe and cue trials of MWM). In the cases where variances were not normal (Distance traveled during NOR phase), Welch's correction was applied to the two-tailed, unpaired t-test. If the normality assumption was violated (Average startle, Average % PPI), the two-tailed Mann-Whitney test was used. For comparing between Habituation and NOR phases in the NOR task, paired, two-tailed t-tests were used. Mixed model 2-way ANOVAs were used to compare the daily consumption of food and weights of the mice in the days after NSF, learning across

days (MWM) and % PPI across different pre-pulse intensities (PPI). Differences were considered to be significant if p < 0.05.

## Results

### Mice show altered activity patterns in the absence of USP2

WT and *Usp2* KO mice were placed in running wheel cages and locomotor activity was recorded under a 12h:12h light:dark cycle (representative actograms in Fig 1A). *Usp2* KO mice showed a trend (p = 0.06) for more total daily activity than WT mice (Fig 1B). Further, the duration of activity bouts was significantly longer in KO mice (p = 0.049, Fig 1C) and the average number of bouts per day was lower (p = 0.14, S1A Fig). The total counts of wheel rotation per bout of activity was also significantly increased in *Usp2* KOs, compared to WTs (p = 0.0481, S1B Fig). Overall, these data indicated increased and less fragmented activity in mice lacking USP2. These data are consistent with those previously obtained in our laboratory [11].

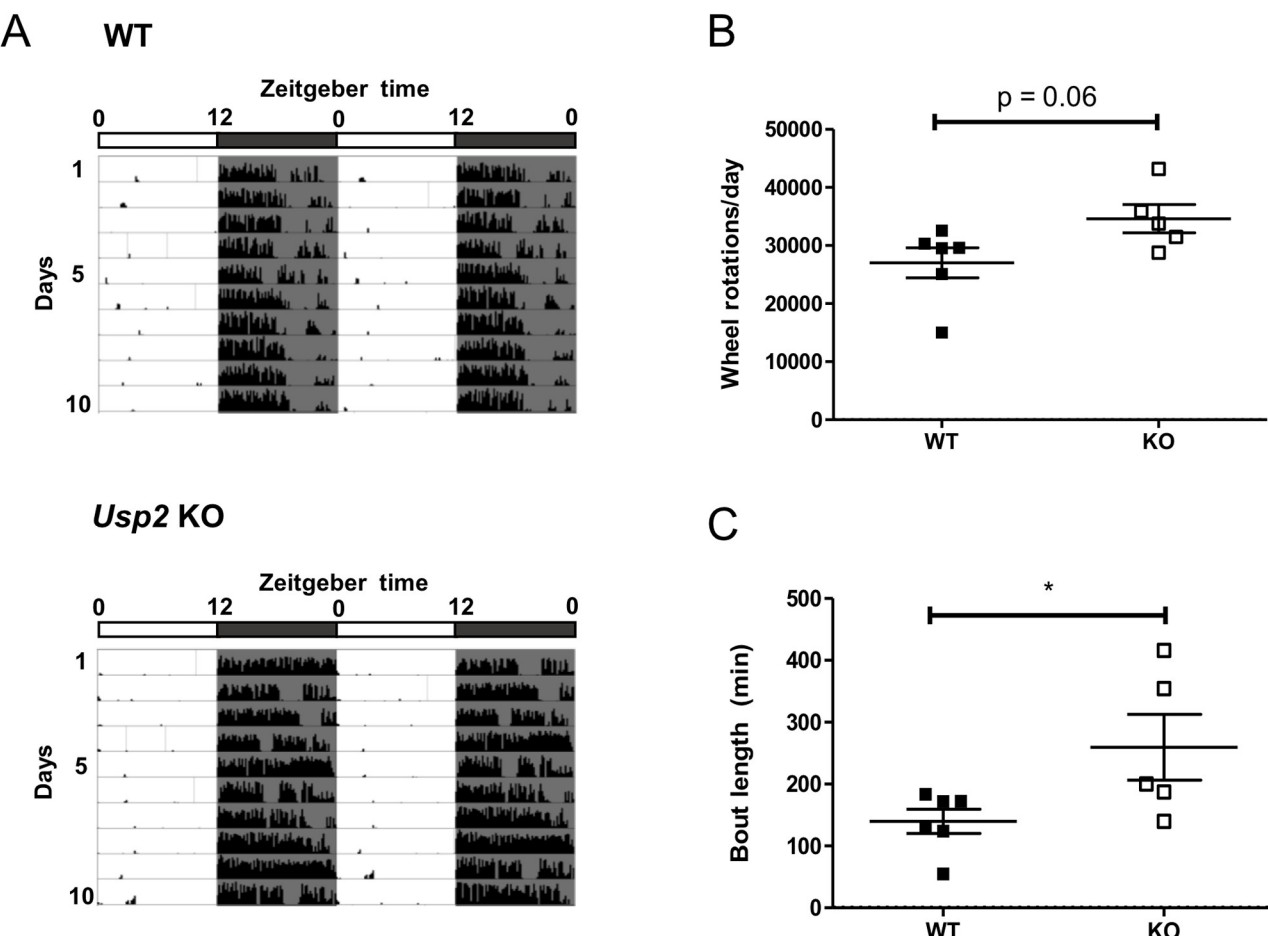

**Fig 1. Increased daily activity and less fragmentation of activity in *Usp2* KO mice.** (A) Representative actograms for the wheel running activity of WT and *Usp2* KO mice over 10 days under a 12h:12h light:dark cycle. (B, C) Quantification, averaged over the 10 days, of the total daily activity (B) and the average bout length (C). Individual data points represent independent mice (n: WT = 6, KO = 5) and data are represented as mean ± SEM. Unpaired two-tailed t-tests, *: p < 0.05.

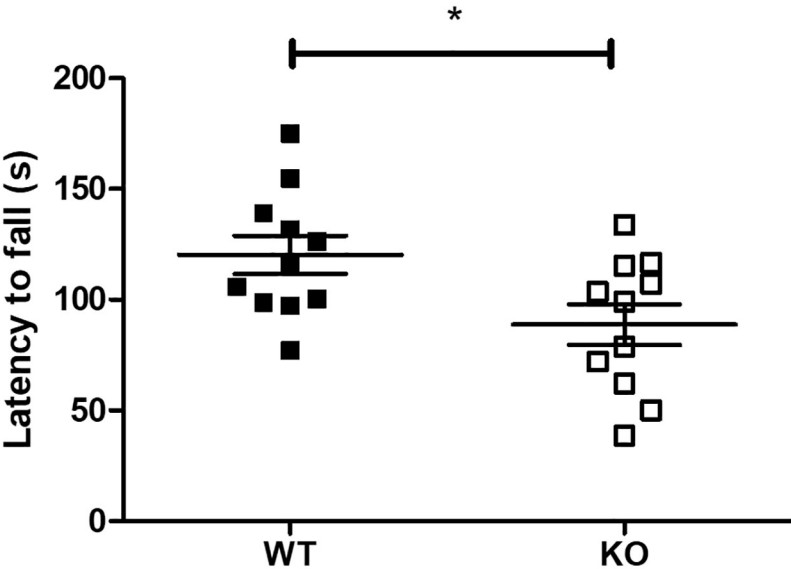

**Fig 2. Reduced motor coordination in *Usp2* KO mice.** Measurement of time spent on the accelerating rotarod by WT and *Usp2* KO mice. Individual data points represent independent mice (n: WT = 11, KO = 11) and data are represented as mean ± SEM. Unpaired two-tailed t-tests, *: p < 0.05.

## Motor coordination is reduced in mice lacking USP2

The increased activity and reduced fragmentation in running wheels led us to wonder if this phenotype could be the result of a change in motor coordination in the *Usp2* KO mice. To assay this, mice were subjected to the rotarod protocol, which assesses motor coordination and balance by evaluating the ability of the mouse to stay on top of a rotating cylinder. *Usp2* KO mice spent significantly less time on the accelerating cylinder (t(20) = 2.49, p = 0.022, Fig 2) and fell from the cylinder at lower speeds (WT: 18.11 ± 1.35 rpm, KO: 13.94 ± 1.31 rpm, t(21) = 2.21, p = 0.038) compared to WT mice, pointing towards a reduction of motor coordination in mice lacking USP2. Hence, a change in motor coordination does not explain the increased activity phenotype seen in the running wheel experiments.

While motivated locomotor activity did not correlate with motor coordination phenotypes, we questioned whether it might have an effect on general locomotion in the *Usp2* KO mice. This was assayed using the open field test (OFT), which assesses the general locomotion of mice when subjected to a novel environment. The total distance covered by the mice in 50 minutes showed no differences between the general locomotion of the two genotypes (t(15) = 0.884, p = 0.391, S1C Fig).

## Anxiety-like behavior is decreased in mice lacking USP2

We surveyed *Usp2* gene expression in the mouse brain expression database in the Allen Mouse Brain Atlas (https://mouse.brain-map.org/experiment/show?id=76098316) [20]. As shown in Table 1, *Usp2* is expressed throughout the brain. Therefore, we subjected the WT and *Usp2* KO mice to a battery of tests for affective and cognitive behaviors corresponding to these different brain regions.

Expression of the gene in regions such as the hippocampus and cortex, which are associated with anxiety-like behavior [33], prompted us to assess such behavior, using the Elevated Plus Maze (EPM) and the Novelty-suppressed Feeding (NSF) test.

**Table 1. Relative expression of *Usp2* transcript in the mouse brain.**

| Brain region | *Usp2* expression level * |
|---|---|
| Isocortex | ++++++ |
| Olfactory bulb | +++ |
| Hippocampus | ++++ |
| Striatum | ++++ |
| Thalamus | +++ |
| Hypothalamus | + |
| Midbrain | + |
| Pons | + |
| Cortical sub-plate | +++++ |
| Pallidum | ++ |
| Medulla | + |
| Cerebellum | ++ |

* Relative expression from in situ hybridization data for *Usp2*, in the Allen mouse brain atlas (each '+' sign represents ~1 unit of raw expression value).

The EPM assay builds on the natural aversion of mice to heights and open spaces, counterbalanced by their drive to explore novel surroundings. When subjected to the EPM, *Usp2* KO mice exhibited a trend towards reduced anxiety-like phenotypes compared to WT mice: they spent a greater proportion of time in the open arms as compared to the WTs (t(15) = 1.79, p = 0.09, Fig 3A). The latency to enter open arms, on the other hand, did not differ significantly between genotypes (WT: 34.67 ± 29.7 s, KO: 18.23 ± 5.98 s, Mann-Whitney U test: U = 20.5, p = 0.2275). Further, the absence of differences in general locomotion between WT and KO mice in the OFT (S1C Fig) confirms that the time spent in the different arms of the EPM is based on the anxiety-like levels of the mice. Thus, the EPM data indicates a reduced anxiety-like phenotype in the *Usp2* KO mice.

The NSF test opposes the desire for safety with the desire to feed. The *Usp2* KO mice started feeding faster than the WT mice (t(15) = 3.323, p = 0.0046, Fig 3B). When feeding was assayed in home cages right after the test, for 10 minutes, all mice ate equally (t(15) = 0.403, p = 0.692, S2A Fig). To verify if the increased latency to feed was a result of metabolic changes, or changes in hunger, mice were weighed daily, and their daily food consumption was measured. There were no differences between WT and KO mice in their general appetite, as both genotypes consumed equal amounts of food over the span of a week (F (8, 120) = 1.63, p = 0.1232, Fig 3C). Similarly, there were no differences in the weights of the mice recorded over a 9-day span (F (4, 60) = 0.1, p = 0.9812, S2B Fig). Thus, the observed phenotype of an increased latency to feed reflects mainly on a decrease in the anxiety-like behavior in mice lacking the *Usp2* gene, consistent with the trend observed in the EPM test results.

## Recognition memory, but not spatial memory, is attenuated in mice lacking USP2

Since *Usp2* is highly expressed in the cortex and hippocampus (Table 1), we questioned whether knocking it out would affect working memory [34]. In the NOR test, WT mice had a significantly higher DR in the NOR phase of the test (t(22) = 3.0, p = 0.007, Fig 4A). On the other hand, for the *Usp2* KO mice, the DR was unchanged between the habituation and NOR phases (t(21) = 0.364, p = 0.72, Fig 4B). Comparison between the discrimination ratios of WT and KO mice showed that the KOs had a significantly lower ability to distinguish novel objects

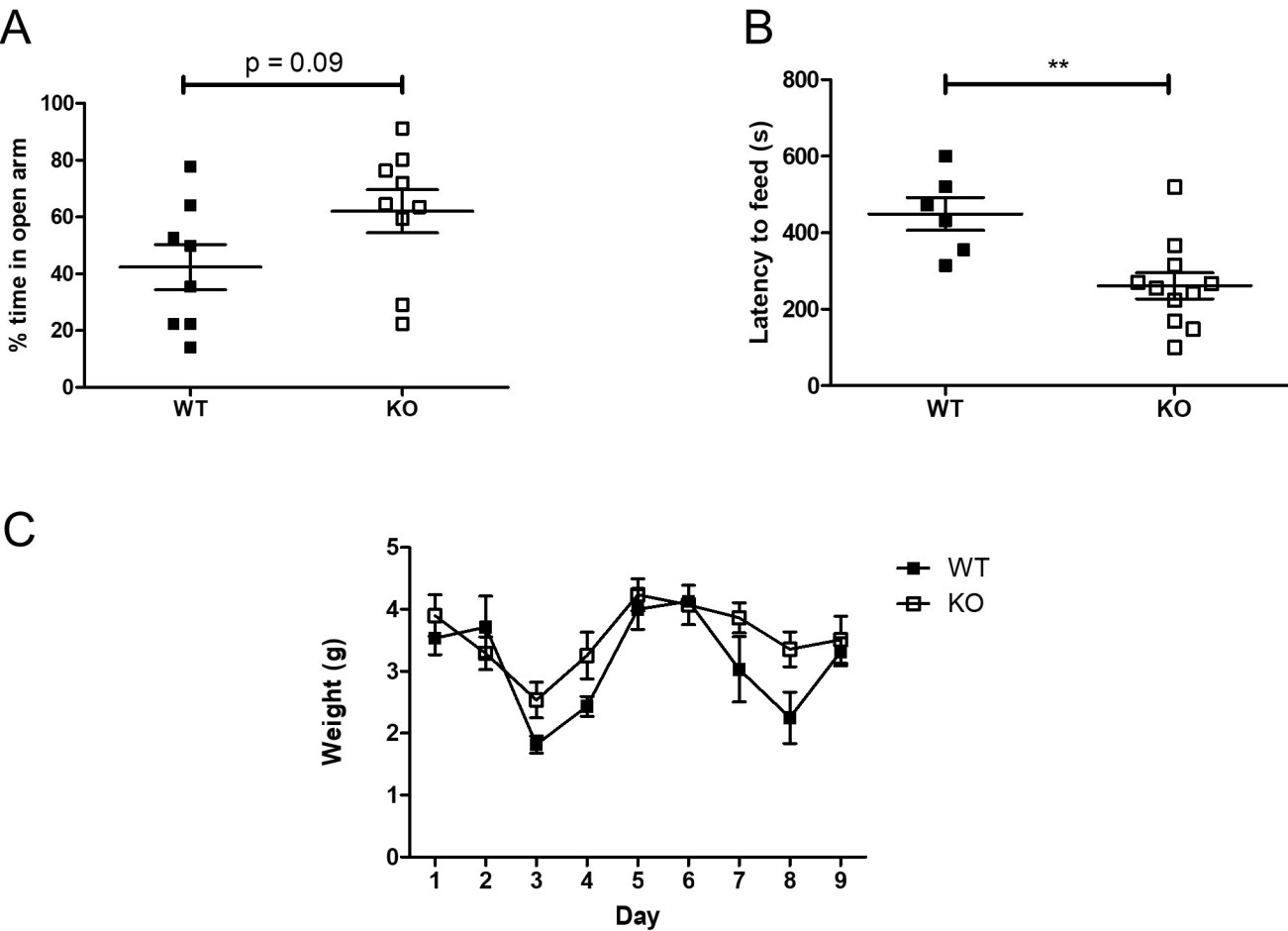

**Fig 3. Reduced anxiety-like behavior in *Usp2* KO mice.** (A) Proportion of time spent in the open arms of the elevated plus maze (EPM). (B) Latency to start feeding in the testing arena of the novelty-suppressed feeding (NSF) test. (C) Food consumed over 9 days following the NSF test. Individual data points represent independent mice (EPM, n: WT = 8, KO = 9; NSF, n: WT = 6, KO = 11) and data are represented as mean ± SEM. Two-way ANOVA (C) or unpaired two-tailed t-tests (A, B, D), **: p < 0.01.

(t(22) = 2.45, p = 0.023, Fig 4C). Therefore, recognition memory is impaired in mice lacking *Usp2*.

Further, to ascertain that the differences in DR do not stem from a change in locomotor activity of the mice, we quantified the average total distance covered by WT and KO mice in the NOR phase of the test. We found no difference in the total locomotion of the two genotypes (Welch corrected t(15) = 0.341, p = 0.738, Fig 4D). This is in line with the results of the OFT test (S1C Fig).

We also tested for spatial memory using the Morris water maze (MWM). Learning was observed in all the mice, over the first 4 days (F (3, 45) = 12.71, p < 0.0001) (Fig 5A). In the probe trial, both genotypes of mice spent more than 25% of the test time in the appropriate quadrant (WT: t(8) = 2.55, p = 0.051, KO: t(10) = 2.17, p = 0.055, Fig 5B), suggesting that the mice had learnt the position of the platform. In the cue trial, all the mice were equally adept at finding the platform at a new location (t(15) = 0.09, p = 0.9235, Fig 5C), confirming that performance factors unrelated to place learning were not involved in the probe trial results. However, in all these procedures, no differences were found between the WT and *Usp2* KO mice.

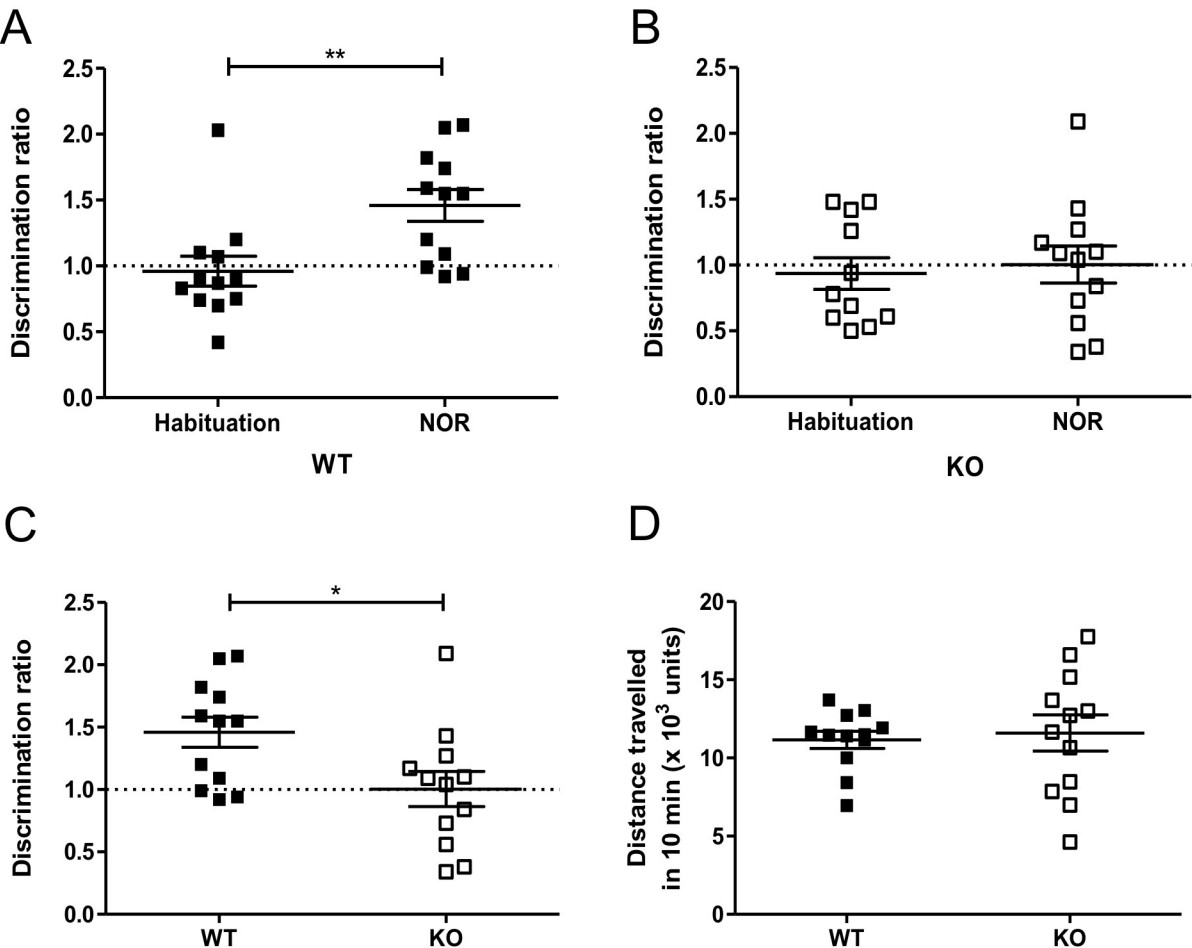

**Fig 4. Impaired object recognition learning and memory in *Usp2* KO mice.** (A, B) Proportion of time spent by WT mice (A) or *Usp2* KO mice (B) to explore the novel object compared to the familiar object (discrimination ratio). (C) Comparison of the discrimination ratios of the WT and KO mice during the novel object recognition phase. (D) Total distance travelled by the WT and KO mice during the NOR phase of the test. Individual data points represent independent mice (n: WT = 12, KO = 12) and data are represented as mean ± SEM. Paired two-tailed t-tests (A, B), unpaired two-tailed t-tests (C) or unpaired two-tailed t-tests with Welch's correction (D), **: p < 0.01, *: p < 0.05.

### Effects on sensorimotor gating in mice lacking USP2

PPI is known to be associated with the dopaminergic system, the limbic system, the olfactory bulb and the prefrontal cortex [35–38]. Among these regions, *Usp2* expression is high in the olfactory bulb, the striatum, the thalamus and the prefrontal cortex (Table 1, [20]). Hence, we tested for pre-pulse inhibition (PPI) of acoustic startle in *Usp2* KO mice. There was no difference between the genotypes in their baseline startle response to a 120 dB startle pulse (t(20) = 0.45, p = 0.661, Fig 6A). The % PPI averaged across the different pre-pulse intensities were also not different between the genotypes (Mann-Whitney U = 41.0, p = 0.212, Fig 6B). However, when the % PPI data were analyzed separately for each pre-pulse intensity, a Genotype x Pre-pulse intensity interaction was seen (F (3, 63) = 2.79, p = 0.043, Fig 6C). A simple main effect analysis revealed a trend for a difference between genotypes for a 12 dB pre-pulse intensity (p = 0.06). This suggested that *Usp2* may affect sensorimotor gating.

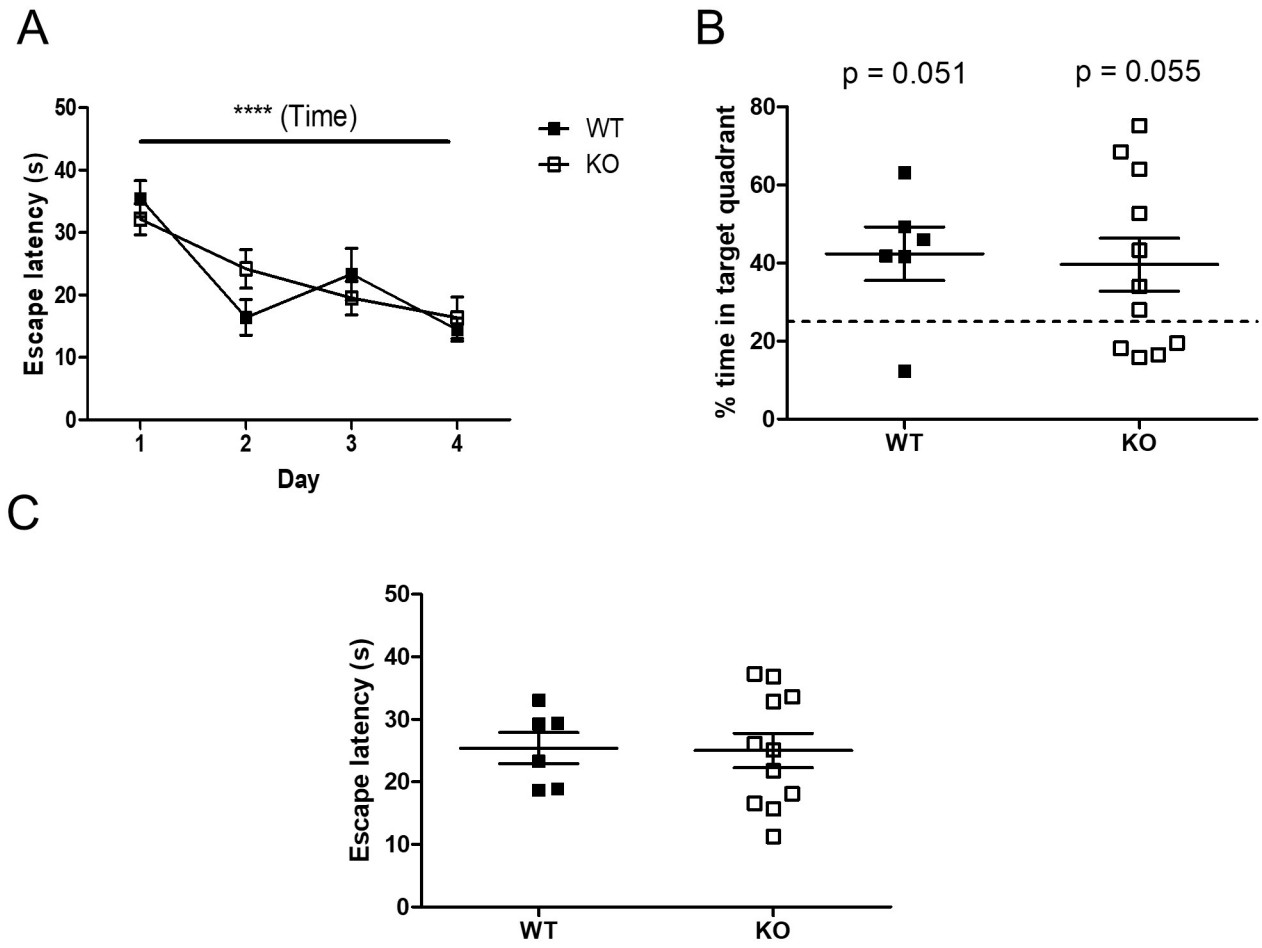

**Fig 5. Unaltered spatial learning and memory of *Usp2* KO mice.** (A) Time taken by the mice to find the platform on subsequent days, as a proxy for learning. (B) Probe trial: Proportion of time spent in the target quadrant in the absence of the platform on day 5. (C) Cue trial: Time taken by the mice to find the platform at a new location. Individual data points represent independent mice (n: WT = 6, KO = 11) and data are represented as mean ± SEM. Two-way ANOVA (A) or unpaired two-tailed t-tests and one sample t-test (B, C), ****: $p < 0.0001$ for effect of time in (A), interaction and effect of genotype n.s.

## Discussion

In this report, we present evidence for a role of the deubiquitinating enzyme USP2 in the central nervous system. More specifically, we have uncovered alterations of anxiety-like behavior, learning and memory, and motor coordination in mice with a deletion in the *Usp2* gene. This work represents, to our knowledge, the first characterization of behaviors in the absence of USP2 function, beyond its established role in the regulation of circadian rhythms.

USP2 is a well-studied DUB with multiple established functions. The first known substrate of USP2 was the fatty acid synthase protein (FAS). In 2004, Graner and colleagues showed that USP2 binds to and stabilizes FAS [39], a protein known to be upregulated in many cancers including prostate cancer [40]. USP2 was shown to be involved in regulating the degradation of oncogene p53 by targeting the ubiquitin ligase Mdm2 [41]. MdmX, another target of Mdm2, is also deubiquitinated by USP2. Further, USP2 controls the cell cycle by deubiquitinating Aurora-A, a centrosomal kinase required for mitosis [42], as well as cyclins such as cyclin A1 and cyclin D [43–45], which are regulators of meiosis. Hence, unsurprisingly, the dysregulation of USP2 levels was associated with the development of various kinds of cancers, such as

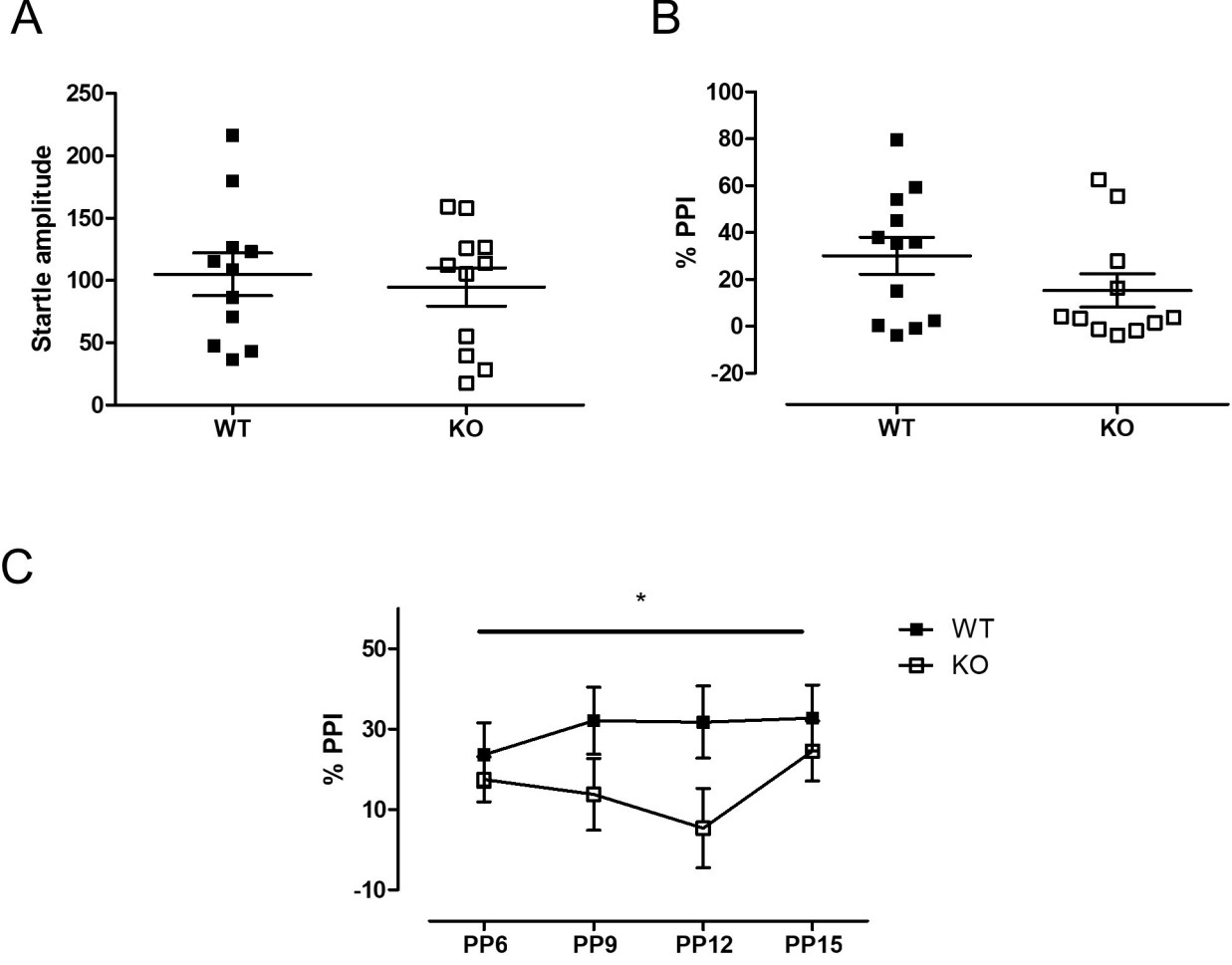

**Fig 6. Reduced sensorimotor gating in *Usp2* KO mice.** (A) Amplitude of baseline startle when a 120 dB pulse is given in the absence of any pre-pulse. (B) Average Percent pre-pulse inhibition (% PPI) of all of the non-zero pre-pulse intensities administered. (C) % PPI in response to each of the different intensities of pre-pulses administered. In (A) and (B), individual data points represent independent mice (n: WT = 11, KO = 11) and data are represented as mean ± SEM. In (C), each data point represents the average %PPI for the corresponding PPI. Two-way ANOVA (C) or two-tailed Mann-Whitney test (A, B), *: p < 0.05 for the interaction in (C).

colorectal cancer [44], prostate cancer [43] and oral squamous cell carcinoma [46]. In the circadian system, USP2 regulates the ubiquitination of several clock proteins and contributes to the response of the circadian clock to light [11–14]. Apart from its functions within the clock, USP2 also mediates clock output. For instance, USP2 regulates the membrane scaffolding protein NHERF, in a circadian manner [47]. NHERF, in turn, regulates cellular homeostasis of calcium absorption in a clock-dependent manner [47], hence making USP2 a clock output mediator.

While several functions of USP2 have been uncovered, little is known about its possible roles in the central nervous system. Based on data from the Allen Mouse Brain Atlas, *Usp2* gene is expressed in various brain regions (Table 1), which suggests that this DUB could be regulating behavioral processes. We addressed this using a battery of tests aimed at assaying various behaviors. This was initially prompted by our analyses of wheel-running behavior, in which we noted increased daily activity and a more consolidated activity pattern in *Usp2* KO mice. The reduction in activity fragmentation could be the result of an effect of the *Usp2* KO on the circadian system but our prior work using constant conditions, where limited effects on

the endogenous period of the rhythms were found, argues against this. Thus, we used the rotarod to check a possible impact of *Usp2* gene deletion on motor coordination: the decrease in motor coordination of the KO mice suggests that this is also not the source of the altered activity patterns in mice lacking *Usp2*. Another possible source for this phenotype could be in the light-response pathways; however, the increased activity and consolidation is seen not only in the light phase of the 12:12LD cycle but also in the night, which suggests that an alteration of the light-response pathways (e.g. reduced masking of activity by light) is not involved. Finally, running wheel activity being a motivated behavior [48, 49], the phenotype in these assays could be due to changes in the reward pathways in the KO mice [50, 51]. Indeed, *Usp2* is highly expressed in the striatum (Table 1). Therefore, although we have not tested the *Usp2* KO mice for reward behaviors, this could represent an interesting avenue for future research.

The hippocampus and the cortex show the highest levels of *Usp2* expression. These are regions associated with the control of anxiety-like behavior. Accordingly, we found a decrease in anxiety-like behavior in mice lacking *Usp2*, both in the EPM and the NSF tests. In the former, the mice lacking *Usp2* spent more time in the open arms than their WT littermates, showing a reduction in anxiety-like behavior. In line with these findings, in the NSF test (which is a more sensitive measure of anxiety-like behavior, due to the pressure to feed), the mice lacking *Usp2* fed significantly quicker than the WT mice, reiterating the reduction in anxiety-like behavior in mice lacking USP2.

Further exploring the effects of cortical and hippocampal expression of *Usp2*, we looked at the effects its deletion might have on cognition. In the NOR test, the KOs showed no distinction in interacting with the novel and the familiar object (DR of 1 in both, the habituation and NOR phases), pointing to a deficit in either the learning of the two objects or to a deficit in working memory. On the other hand, the MWM test showed that spatial learning and memory were intact in the *Usp2* KO mice. Interestingly, a prior study has shown a correlation between a reduction of spatial memory in rats following stress with a downregulation of USP2 protein levels in the hippocampus, both reductions being concomitantly rescued by treatment with retigabine, an opener of Kv7 potassium channels [52]. This suggested that USP2 might play a role in spatial memory, at least under stress conditions, in rats. On the contrary, our data indicate no role for USP2 in spatial memory in mice under basal, unchallenged conditions.

Finally, using the PPI test, we looked at sensorimotor gating, or the ability of sensory inputs to guide motor responses in the organism. This behavior is guided by the striatum and olfactory bulbs, where the expression of *Usp2* is high as well. Also, PPI has been known to be affected by the limbic and dopaminergic systems: it is especially affected by the dopaminergic cortico-striato-pallido-thalamic and limbic cortico-striato-pallido-pontine tracts [35]. Most of these regions also have a high expression of *Usp2*. Surprisingly, rather mild phenotypes were found in the PPI test. There seemed to be an effect of genotype on the startle response, but not at all pre-pulse intensities. Thus, further work will be required to delineate the effects of USP2 on sensorimotor gating.

In conclusion, our data indicate that *Usp2* plays a role not only in the circadian system, but also in controlling various other behaviors such as anxiety-like phenotypes, motor coordination and working memory. A limitation of our study is that it does not provide insights on what might be the substrates of USP2 in the brain regions where it is highly expressed. For the same reason, it is hard to speculate on whether the various phenotypes of the *Usp2* KO mice are due to shared vs. separate mechanisms within the brain. It will be an important future direction to find out what these substrates are, and how USP2 action on these proteins can lead to behavioral alterations like the ones we have observed in this study. Nevertheless, this report sets the stage for the study of the roles of USP2 in the central nervous system and shows that this deubiquitinase has many roles beyond those already described in peripheral organs.

## Supporting information

**S1 Fig. Locomotor activity patterns of *Usp2* KO mice.** (A, B) Quantification of averaged wheel running activity over 10 days: number of bouts of activity per day (A) and number of wheel rotations per activity bout (B). (C) Open field measurement of total distance traveled by WT and *Usp2* KO mice as a measure of general locomotion. Individual data points represent independent mice (Wheel-running activity, n: WT = 6, KO = 5; Actimetry, n: WT = 6, KO = 11) and data are represented as mean ± SEM. Unpaired two-tailed t-tests, * p < 0.05.
(TIF)

**S2 Fig. Control measures of novelty-suppressed feeding (NSF) test.** (A) Quantity of food consumed within 10 minutes post-NSF test, in the home cage. (B) Weight of the mice over 9 days following the NSF test. Individual data points represent independent mice (n: WT = 6, KO = 11) and data are represented as mean ± SEM. Unpaired two-tailed t-tests (A) or two-way ANOVA (B), all n.s.
(TIF)

## Acknowledgments

We thank all the members of the Cermakian lab for discussions. We thank Tara Delorme, Marie-Eve Cloutier, Geneviève Dubeau Laramée, Christine Kirady and Adeline Rachalski for technical help and for assistance in the behavioral experiments; Joseph Rochford, Lalit K. Srivastava and Eve-Marie Charbonneau for advice on the behavioral tests; Joseph Rochford for a critical review of the manuscript; Yu Ding and William Ozell-Landry for data analysis; Simon S. Wing and Nathalie Bédard for advice and technical help with the *Usp2* KO mice.

## Author Contributions

**Conceptualization:** Nicolas Cermakian.

**Data curation:** Shashank Bangalore Srikanta, Katarina Stojkovic.

**Formal analysis:** Shashank Bangalore Srikanta.

**Funding acquisition:** Nicolas Cermakian.

**Investigation:** Shashank Bangalore Srikanta, Katarina Stojkovic.

**Methodology:** Shashank Bangalore Srikanta, Katarina Stojkovic.

**Project administration:** Nicolas Cermakian.

**Supervision:** Nicolas Cermakian.

**Validation:** Shashank Bangalore Srikanta.

**Visualization:** Shashank Bangalore Srikanta.

**Writing – original draft:** Shashank Bangalore Srikanta.

**Writing – review & editing:** Katarina Stojkovic, Nicolas Cermakian.

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
