## [Decision Letter · Decision Letter 0]

27 Nov 2020

PONE-D-20-32107

Behavioral phenotyping of mice lacking the deubiquitinase USP2

PLOS ONE

Dear Dr. Cermakian,

Thank you for submitting your manuscript to PLOS ONE. After careful consideration, we feel that it has merit but does not fully meet PLOS ONE’s publication criteria as it currently stands. Therefore, we invite you to submit a revised version of the manuscript that addresses the points raised during the review process.

Provide a clearer rationale for the new tests in the introduction.

Consider providing an alternative measure of locomotor activity.

Consider re-analyzing the novel object recognition data.

Improve the discussion to include inter-relations between different phenotypes.

Complete statistics section and provide F/df values for stats.

We look forward to receiving your revised manuscript.

Kind regards,

Henrik Oster, Ph.D.

Academic Editor

PLOS ONE

Journal Requirements:

Reviewers' comments:

Reviewer's Responses to Questions

**Comments to the Author**

1. Is the manuscript technically sound, and do the data support the conclusions?

Reviewer #1: Partly

Reviewer #2: Partly

2. Has the statistical analysis been performed appropriately and rigorously? 

Reviewer #1: Yes

Reviewer #2: Yes

3. Have the authors made all data underlying the findings in their manuscript fully available?

Reviewer #1: Yes

Reviewer #2: Yes

4. Is the manuscript presented in an intelligible fashion and written in standard English?

Reviewer #1: Yes

Reviewer #2: Yes

5. Review Comments to the Author

Reviewer #1: Srikanta et al., phenotyped mice lacking deubiquitinase USP2 in various behavioral assays relating to locomotion, anxiety, memory and sensorimotor gating. Though this study is novel and interesting, there are some questions that needs to be addressed to prompt publication.

Major comments

It would good to include another measure of locomotor activity than the running wheel, as baseline locomotor changes could influence your readout in the anxiety assays and novel object recognition assay. For instance, did you compare number of crossings in the EPM?

To be sure that there is no differences in explorative behavior due to locomotor activity changes, it would be relevant to perform a general locomotor measurement in a home-cage-like environment.

As the novel object recognition test failed in the WT group (figure 4A), it unfortunately makes the overall results from the test difficult to interpret. Please revise – perhaps this can be solved with a re-analysis of your data (see point below).

Please re-analyze the novel object recognition data! The criterion for object-exploration of 5 cm distance between mouse and object seems unusually large - 2 cm is a more common criterion for exploration behavior. This may also explain why the WT-group showed no significant increase in discrimination ratio for the novel object in figure 4A.

Overall the manuscript is well written and to the point. However, to reach a clearer overview of the USP2-KO phenotype it is essential to deepen the discussion on the specific behaviors and how these may interrelate. It could be that the USP2-KO phenotype arising from one behavioral assay influences that of the others. For instance, decreased anxiety, could be a result from increased exploration behavior due to locomotion, and could in turn affect novel object interaction; And vice versa deficits in memory-acquisition could influence novelty suppressed feeding. Please revise

Minor comments

Though the title of the manuscript is good, it would be relevant to include information on the specific behavioral findings.

You mention the Forced Swim test in your materials and methods section, line 97 and line 219, but you haven’t included any data from the test nor any further description of it – please revise.

In the discussion of your PPI results, please consider other relevant limbic circuits expressing USP2, such as the thalamocortical circuit.

Reviewer #2: This manuscript describes a series of experiments investigating the behavioural sequalae of USP2 deletion across a range of different cognitive and motoric domains. Previous studies, by the authors, had demonstrated circadian activity pattern differences in these mice, hence the aims to expand the understanding of USP2 deletion across other behaviours. The findings open with a replication of the initial findings before exploring effects across anxiety, motor co-ordination, two types of memory and sensory motor gating – all reasonable target to evaluate. The studies were conducted following proven methodologies and with due consideration of ethical and legal constraints for the use of animals in research. Overall, the manuscript is well presented and clearly written, results are displayed well and the statistical approach is sound. I think that the Introduction could have expanded upon why the particular tests/domains were chosen rather than dwelling quite so much on the previous results. The Discussion section was also well written and presented and dealt with the findings in context but see below for comments on further data evaluation to strengthen the anxiety findings. Likewise, the authors probably have the data that will help to explain away some fo the NOR effects – see below for comments on this.

Overall, the study itself has good merit in its approach and the results, when further investigated, which would be relevance to the field but currently the manuscript and data needs further investigation to underpin the results included.

In working through the manuscript, I have the following major issues:

4hr delay in the NOR is not really short-term memory – researchers would generally drop to the realms of 10min for this – 4hrs is also not really long-term either, but an intermediate time.

Stats section not really including all the analyses performed – not clear what is meant by average %PPI and whether this is helpful, as it is masking the interaction between prepulse intensity and response…

Results section – full stats were missing in the text – only p values given, the t or F and df should be listed.

For the EPM data – the mice were tracked on the apparatus and it would be interested to contrast the decreased anxiety observed with correlations to whether they demonstrated increased activity? The mice are hyperactive in the circadian rhythm tests and maybe on the EPM – so maybe will explore more, and appear less anxious but maybe not so in reality – so a measure of locomotion in the EPM – expressed over the 5min test would be interesting to see and help to determine exactly what is shown in this test. Similarly, the NSF, NOR and MWM could also all be confounded by hyperactivity.. so again a report on these behaviours in these tasks would be interesting to see.

For the PPI – in the methods the authors alerted to increased PPI (pp11, l 211/212), whereas there is stronger evidence for reduced PPI effects being more linked to models of mental health conditions etc – so this pointer here could be rephrased and referenced accordingly – which makes a stronger point to the actual finding of recued PPI in the USB nulls. (Swerdlow et al 2016 Nov;30(11):1072-1081. doi: 10.1177/0269881116661075.)

For the NOR study, reporting just the DR does not give clarity to the results and what they mean – as the authors discuss – actual object contact times will determine if there were issues during the acquisition stage that could affect memory testing later – maybe correlate exposure to later memory testing? Also, would be interesting to see a comparison to chance for each group at tets – looks like the WT maybe sig but the KO not – so might demonstrate memory impairment in the KOs, even though not sig different to WT (see Hall et al 2016 130:118-28. doi: 10.1016/j.nlm.2016.02.002) for this approach.

Minor points.

Use of SI units throughout- not Imperial units

More clarity in final N for each study

P7, L118 – edit rpmrpm

P7, L120 – edit got to was

EPM methods needs to be in past tense

6. PLOS authors have the option to publish the peer review history of their article (what does this mean?). If published, this will include your full peer review and any attached files.

Reviewer #1: No

Reviewer #2: No

---

## [Author Response · Author response to Decision Letter 0]

10 Jan 2021

Response to Reviewers of manuscript PONE-S-20-39745 by Srikanta et al. 

We wish to thank the editor and the reviewers for their comments on the manuscript. As described below, we have addressed all comments and modified the manuscript accordingly.

Editor:

#1: Provide a clearer rationale for the new tests in the introduction.

Answer: We have revised the last paragraph of the Introduction to provide such a rationale:

“Further, based on high Usp2 expression in the hippocampus and cortex, we tested the mice for anxiety-like behavior and memory, and noticed a reduction in anxiety-like phenotypes and deficits in the working memory of Usp2 KO mice, but not in their spatial memory. Based on Usp2 expression in the olfactory bulb, striatum, thalamus and prefontal cortex, we assessed sensorimotor gating, for which limited effects of Usp2 gene deletion were found.” (Page 4, Line 63)

#2: Consider providing an alternative measure of locomotor activity.

Answer: Although we hadn’t included actimetry in the initial manuscript, we have now added these data. They show a lack of differences between WT and KO mice (p = 0.391), in their general locomotion. Further, analysis of total distance covered by the mice during the NOR trial also showed no difference in general locomotion between the WT and KO groups (p = 0.665). Hence, our observations in the various tests are not related to changes in the general locomotion of the two genotypes themselves. We have added a description of these new data in the Results section and show them in new Figures S1C and 4D:

“While motivated locomotor activity did not correlate with motor coordination phenotypes, we questioned whether it might have an effect on general locomotion in the Usp2 KO mice. This was assayed using the open field test (OFT), which assesses the general locomotion of mice when subjected to a novel environment. The total distance covered by the mice in 50 minutes showed no differences between the general locomotion of the two genotypes (t(15) = 0.884, p = 0.391, S1C Fig).” (Page 14, Line 275)

“Further, to ascertain that the differences in DR do not stem from a change in locomotor activity of the mice, we quantified the average total distance covered by WT and KO mice in the NOR phase of the test. We found no difference in the total locomotion of the two genotypes (Welch corrected t(15) = 0.341, p = 0.738, Fig 4D). This is in line with the results of the OFT test (S1C Fig).” (Page 18, Line 345)

#3: Consider re-analyzing the novel object recognition data.

Answer: Based on the inputs from the editor and the reviewers, we reduced the area of tracking around the object to the suggested distance of 2 to 3 cm around the object. Further, we initially tracked using the center of the body, but this time, we tracked the mouse using its nose, to be certain of the interaction of the mouse with the two objects. The comparison of the discrimination ratios of the WT mice in the habituation and NOR conditions is now significant (p = 0.007), confirming that they are able to discriminate between the objects and validating the test. The manuscript has been updated accordingly (revised Methods, revised Results, revised Figure 4A - C):

“A mouse was considered to be involved in object exploration if its head was directed towards the object, within a radius of approximately 2 to 3 cm from it.” (Page 9, Line 180 revised text is underlined)

“In the NOR test, WT mice had a significantly higher DR in the NOR phase of the test (t(22) = 3.0, p = 0.0078, Fig 4A). On the other hand, for the Usp2 KO mice, the DR is unchanged between the habituation and NOR phases (t(21) = 0.364, p = 0.72916, Fig 4B). Comparison between the discrimination ratios of WT and KO mice showed that the KOs had a significantly lower ability to distinguish novel objects (t(22) = 2.45, p = 0.0232, Fig 4C). Therefore, recognition memory is impaired in mice lacking Usp2.” (Page 17, Line 329, revised text is underlined)

#4: Improve the discussion to include inter-relations between different phenotypes.

Answer: We prefer to avoid overinterpreting the data: many correlations between behavioral measures can be found in the literature, but they aren’t necessarily causal, since other reports exist in literature supporting the opposite correlations as well. In the manuscript, we have added a sentence in the last paragraph of Discussion to state this:

“For the same reason, it is hard to speculate on whether the various phenotypes of the Usp2 KO mice are due to shared vs. separate mechanisms within the brain.” (Page 23, Line 457)

#5: Complete statistics section and provide F/df values for stats.

Answer: F, df and p-values were already provided in the original manuscript, where applicable. We have now also included the t-value with their respective dfs, for all the t-tests.

Reviewer 1:

#1: It would good to include another measure of locomotor activity than the running wheel, as baseline locomotor changes could influence your readout in the anxiety assays and novel object recognition assay. For instance, did you compare number of crossings in the EPM?

To be sure that there is no differences in explorative behavior due to locomotor activity changes, it would be relevant to perform a general locomotor measurement in a home-cage-like environment.

Answer: Although we hadn’t included actimetry in the initial manuscript, we have now added these data. They show a lack of differences between WT and KO mice (p = 0.391), in their general locomotion. Further, analysis of total distance covered by the mice during the NOR trial also showed no difference in general locomotion between the WT and KO groups (p = 0.665). Hence, our observations in the various tests are not related to changes in the general locomotion of the two genotypes themselves. We have added a description of these new data in the Results section and show them in new Figure S1C and 4D:

“While motivated locomotor activity did not correlate with motor coordination phenotypes, we questioned whether it might have an effect on general locomotion in the Usp2 KO mice. This was assayed using the open field test (OFT), which assesses the general locomotion of mice when subjected to a novel environment. The total distance covered by the mice in 50 minutes showed no differences between the general locomotion of the two genotypes (t(15) = 0.884, p = 0.391, S1C Fig).” (Page 14, Line 275)

“Further, to ascertain that the differences in DR do not stem from a change in locomotor activity of the mice, we quantified the average total distance covered by WT and KO mice in the NOR phase of the test. We found no difference in the total locomotion of the two genotypes (Welch corrected t(15) = 0.341, p = 0.738, Fig 4D). This is in line with the results of the OFT test (S1C Fig).” (Page 18, Line 345)

#2: As the novel object recognition test failed in the WT group (figure 4A), it unfortunately makes the overall results from the test difficult to interpret. Please revise – perhaps this can be solved with a re-analysis of your data (see point below).

Please re-analyze the novel object recognition data! The criterion for object-exploration of 5 cm distance between mouse and object seems unusually large - 2 cm is a more common criterion for exploration behavior. This may also explain why the WT-group showed no significant increase in discrimination ratio for the novel object in figure 4A.

Answer: Based on this comment, we reduced the area of tracking around the object to the suggested distance of 2 to 3 cm around the object. Further, we initially tracked using the center of the body, but this time, we tracked the mouse using its nose, to be certain of the interaction of the mouse with the two objects. The comparison of the discrimination ratios of the WT mice in the habituation and NOR conditions is now significant (p = 0.007), confirming that they are able to discriminate between the objects and validating the test. The manuscript has been updated accordingly (revised Methods, revised Results, revised Figure 4A - C):

“A mouse was considered to be involved in object exploration if its head was directed towards the object, within a radius of approximately 2 to 3 cm from it.” (Page 9, Line 180 revised text is underlined)

“In the NOR test, WT mice had a significantly higher DR in the NOR phase of the test (t(22) = 3.0, p = 0.0078, Fig 4A). On the other hand, for the Usp2 KO mice, the DR is unchanged between the habituation and NOR phases (t(21) = 0.364, p = 0.72916, Fig 4B). Comparison between the discrimination ratios of WT and KO mice showed that the KOs had a significantly lower ability to distinguish novel objects (t(22) = 2.45, p = 0.0232, Fig 4C). Therefore, recognition memory is impaired in mice lacking Usp2.” (Page 17, Line 329, revised text is underlined)

#3: Overall the manuscript is well written and to the point. However, to reach a clearer overview of the USP2-KO phenotype it is essential to deepen the discussion on the specific behaviors and how these may interrelate. It could be that the USP2-KO phenotype arising from one behavioral assay influences that of the others. For instance, decreased anxiety, could be a result from increased exploration behavior due to locomotion, and could in turn affect novel object interaction; And vice versa deficits in memory-acquisition could influence novelty suppressed feeding. Please revise.

Answer: We prefer to avoid overinterpreting the data: many correlations between behavioral measures can be found in the literature, but they aren’t necessarily causal, since other reports exist in literature supporting the opposite correlations as well. In the manuscript, we have added a sentence in the last paragraph of Discussion to state this:

“For the same reason, it is hard to speculate on whether the various phenotypes of the Usp2 KO mice are due to shared vs. separate mechanisms within the brain.” (Page 23, Line 457)

#4: Though the title of the manuscript is good, it would be relevant to include information on the specific behavioral findings.

Answer: Given that there are many different phenotypes, the title would become too complex if phenotype details are included. Hence, we would prefer to stick to this general title that we have provided. Since this is the first report of non-circadian behavioural changes in USP2 KO mice, it is appropriate to have such a general title.

#5: You mention the Forced Swim test in your materials and methods section, line 97 and line 219, but you haven’t included any data from the test nor any further description of it – please revise.

Answer: Thank you for bringing this error to our attention. It has been corrected.

#6: In the discussion of your PPI results, please consider other relevant limbic circuits expressing USP2, such as the thalamocortical circuit.

Answer: We have added text about this in the Results and Discussion sections:

“PPI is known to be associated with the dopaminergic and limbic systems [35–38]. Among these regions Usp2 expression is high in the olfactory bulb, striatum, thalamus and prefrontal cortex, which are associated with sensorimotor gating [35–37].” (Page 19, Line 367)

“Also, PPI has been known to be affected by the limbic and dopaminergic systems: it is especially affected by the dopaminergic cortico-striato-pallido-thalamic and limbic cortico-striato-pallido-pontine tracts [35]. Most of these regions also have a high expression of USP2. Surprisingly, proportionally mild phenotypes were found in the PPI test.” (Page 22, Line 447)

Reviewer 2:

#1: 4hr delay in the NOR is not really short-term memory – researchers would generally drop to the realms of 10min for this – 4hrs is also not really long-term either, but an intermediate time.

Answer: The phrase “short-term memory” has been replaced with “working memory” to refer to memory store required for memory recall in such a task.

#2: Not clear what is meant by average %PPI and whether this is helpful, as it is masking the interaction between pre-pulse intensity and response

Answer: The average %PPI is the average of the %PPI values, across the different PP intensities. This is a measure of overall difference in sensorimotor gating. This measure provides an overview about the capacity for sensorimotor gating in an animal, as opposed to a plot of %PPI for the different PP intensities, which provides a closer look at the differences in the sensorimotor gating capacities of the animals to increasing intensities of prepulses. The legend of Fig 6 was clarified accordingly:

“(B) Average Percent pre-pulse inhibition (% PPI) of all of the non-zero intensities of pre-pulses administered. (C) % PPI in response to each of the different intensities of pre-pulses administered.” (Page 19, Line 379, revised text is underlined)

#3: Results section – full stats were missing in the text – only p values given, the t or F and df should be listed.

Answer: F-values and df had been provided for all the 2-way ANOVAs in the original manuscript (within the manuscript text). For all the t-tests, the p-values had been provided; t-values have now been added with their respective dfs.

#4: For the EPM data – the mice were tracked on the apparatus and it would be interested to contrast the decreased anxiety observed with correlations to whether they demonstrated increased activity? The mice are hyperactive in the circadian rhythm tests and maybe on the EPM – so maybe will explore more, and appear less anxious but maybe not so in reality – so a measure of locomotion in the EPM – expressed over the 5min test would be interesting to see and help to determine exactly what is shown in this test. Similarly, the NSF, NOR and MWM could also all be confounded by hyperactivity. So again, a report on these behaviours in these tasks would be interesting to see.

Answer: To distinguish the hyperactive phenotype in the wheel (a motivated behavior) from general locomotion, we have now provided actimetry data in the manuscript (new Figure S1C). It shows that there are no differences in general locomotion between WT and KO mice (p = 0.391). Hence, we can be sure that the phenotypic differences between the genotypes found in the EPM, NSF or NOR do not stem from variation in the general locomotion of the mice. The Results section was revised accordingly:

“While motivated locomotor activity did not correlate with motor coordination phenotypes, we questioned whether it might have an effect on general locomotion in the Usp2 KO mice. This was assayed using the open field test (OFT), which assesses the general locomotion of mice when subjected to a novel environment. The total distance covered by the mice in 50 minutes showed no differences between the general locomotion of the two genotypes (t(15) = 0.884, p = 0.391, S1C Fig).” (Page 14, Line 275)

“Further, to ascertain that the differences in DR do not stem from a change in locomotor activity of the mice, we quantified the average total distance covered by WT and KO mice in the NOR phase of the test. We found no difference in the total locomotion of the two genotypes (Welch corrected t(15) = 0.341, p = 0.738, Fig 4D). This is in line with the results of the OFT test (S1C Fig).” (Page 18, Line 345)

#5: For the PPI – in the methods the authors alerted to increased PPI (pp11, l 211/212), whereas there is stronger evidence for reduced PPI effects being more linked to models of mental health conditions etc – so this pointer here could be rephrased and referenced accordingly – which makes a stronger point to the actual finding of recued PPI in the USB nulls. (Swerdlow et al 2016 Nov;30(11):1072-1081. doi: 10.1177/0269881116661075.)

Answer: We agree with the reviewer that usually, increased PPI is associated with mental health conditions, e.g. schizophrenia. However, in this manuscript, we are not presenting the Usp2 KO mice as a model for psychiatric diseases. Our aim is simply to characterize various behaviours of these KO mice. In the manuscript, we have been careful to stick to the description of the behavioral data, and we have refrained from drawing implications for mental health.

#6: For the NOR study, reporting just the DR does not give clarity to the results and what they mean – as the authors discuss – actual object contact times will determine if there were issues during the acquisition stage that could affect memory testing later – maybe correlate exposure to later memory testing? Also, would be interesting to see a comparison to chance for each group at tets – looks like the WT maybe sig but the KO not – so might demonstrate memory impairment in the KOs, even though not sig different to WT (see Hall et al 2016 130:118-28. doi: 10.1016/j.nlm.2016.02.002) for this approach.

Answer: In the NOR, the exploration of the novel object is a measure of the extent to which the animal remembers the previous encounter with the familiar object. Thus, this ability to discriminate between the novel and familiar object was measured by the Discrimination Ratio. We reanalysed all the NOR data. We reduced the area of tracking around the object to the suggested distance of 2 to 3 cm around the object (to account for sniffing behavior). Further, we initially tracked using the center of the body, but this time, we tracked the mouse using its nose, to be certain of the interaction of the mouse with the two objects. The comparison of the discrimination ratios of the WT mice in the habituation and NOR conditions is now significant (p = 0.007), confirming what the reviewer says, i.e. that learning has indeed occurred in the WT mice (thus validating the test) but not in the KO mice (hence our conclusion of a working memory impairment phenotype). Also, we now provide new data for the total distance covered by the mice during the NOR trial: the absence of difference between WT and KO confirms that the difference in DR are not due to a change in general activity. The manuscript has been updated accordingly (revised Methods, revised Results, revised Figure 4):

“A mouse was considered to be involved in object exploration if its head was directed towards the object, within a radius of approximately 2 to 3 cm from it.” (Page 9, Line 180 revised text is underlined)

“In the NOR test, WT mice had a significantly higher DR in the NOR phase of the test (t(22) = 3.0, p = 0.0078, Fig 4A). On the other hand, for the Usp2 KO mice, the DR is unchanged between the habituation and NOR phases (t(21) = 0.364, p = 0.72916, Fig 4B). Comparison between the discrimination ratios of WT and KO mice showed that the KOs had a significantly lower ability to distinguish novel objects (t(22) = 2.45, p = 0.0232, Fig 4C). Therefore, recognition memory is impaired in mice lacking Usp2.” (Page 17, Line 329, revised text is underlined)

“Further, to ascertain that the differences in DR do not stem from a change in locomotor activity of the mice, we quantified the average total distance covered by WT and KO mice in the NOR phase of the test. We found no difference in the total locomotion of the two genotypes (Welch corrected t(15) = 0.341, p = 0.738, Fig 4D). This is in line with the results of the OFT test (S1C Fig).” (Page 18, Line 345)

#7: Use of SI units throughout- not Imperial units

Answer: These have been revised.

#8: More clarity in final N for each study

Answer: We have provided the exact mouse numbers per group for each graph in each figure legend.

#8: P7, L118 – edit rpmrpm

Answer: It is acceleration and hence, the units are rotations per minute per minute (rpmpm)

#9: P7, L120 – edit got to was

Answer: Revised as suggested.

#10: EPM methods needs to be in past tense

Answer: Revised as suggested.

---

## [Decision Letter · Decision Letter 1]

9 Feb 2021

Behavioral phenotyping of mice lacking the deubiquitinase USP2

PONE-D-20-32107R1

Dear Dr. Cermakian, dear Nico,

We’re pleased to inform you that your manuscript has been judged scientifically suitable for publication and will be formally accepted for publication once it meets all outstanding technical requirements.

Kind regards,

Henrik Oster, Ph.D.

Academic Editor

PLOS ONE

Additional Editor Comments (optional):

Reviewers' comments:

Reviewer's Responses to Questions

**Comments to the Author**

1. If the authors have adequately addressed your comments raised in a previous round of review and you feel that this manuscript is now acceptable for publication, you may indicate that here to bypass the “Comments to the Author” section, enter your conflict of interest statement in the “Confidential to Editor” section, and submit your "Accept" recommendation.

Reviewer #1: All comments have been addressed

2. Is the manuscript technically sound, and do the data support the conclusions?

Reviewer #1: Yes

3. Has the statistical analysis been performed appropriately and rigorously? 

Reviewer #1: Yes

4. Have the authors made all data underlying the findings in their manuscript fully available?

Reviewer #1: Yes

5. Is the manuscript presented in an intelligible fashion and written in standard English?

Reviewer #1: Yes

6. Review Comments to the Author

Reviewer #1: The authors have addressed my questions satisfactorily. Overall the manuscript is suited for publication.

7. PLOS authors have the option to publish the peer review history of their article (what does this mean?). If published, this will include your full peer review and any attached files.

Reviewer #1: No

---

## [Editor Report · Acceptance letter]

15 Feb 2021

PONE-D-20-32107R1 

Behavioral phenotyping of mice lacking the deubiquitinase USP2 

Dear Dr. Cermakian:

I'm pleased to inform you that your manuscript has been deemed suitable for publication in PLOS ONE. Congratulations! Your manuscript is now with our production department. 

Kind regards, 

on behalf of

Prof. Henrik Oster 

Academic Editor

PLOS ONE